# Cross-Comparison between Sun-Synchronized and Geostationary Satellite-Derived Land Surface Temperature: A Case Study in Hong Kong

Ibrahim Ademola Adeniran [1], Rui Zhu [1], Jinxin Yang [2], Xiaolin Zhu [1] and Man Sing Wong [1,2,*]

[1] Department of Land Surveying and Geo-Informatics, The Hong Kong Polytechnic University, Hong Kong, China
[2] School of Geography and Remote Sensing, Guangzhou University, Guangzhou 510006, China
* Correspondence: ls.charles@polyu.edu.hk

**Abstract:** Harmonization of satellite imagery provides a good opportunity for studying land surface temperature (LST) as well as the urban heat island effect. However, it is challenging to use the harmonized data for the study of LST due to the systematic bias between the LSTs from different satellites, which is highly influenced by sensor differences and the compatibility of LST retrieval algorithms. To fill this research gap, this study proposes the comparison of different LST images retrieved from various satellites that focus on Hong Kong, China, by applying diverse retrieval algorithms. LST images generated from Landsat-8 using the mono-window algorithm ($MWA_{L8}$) and split-window algorithm ($SWA_{L8}$) would be compared with the LST estimations from Sentinel-3 SLSTR and Himawari-8 using the split-window algorithm ($SWA_{S3}$ and $SWA_{H8}$). Intercomparison will also be performed through segregated groups of different land use classes both during the daytime and nighttime. Results indicate that there is a significant difference among the quantitative distribution of the LST data generated from these three satellites, with average bias of up to $-1.80$ K when $SWA_{H8}$ was compared with $MWA_{L8}$, despite having similar spatial patterns of the LST images. The findings also suggest that retrieval algorithms and the dominant land use class in the study area would affect the accuracy of image-fusion techniques. The results from the day and nighttime comparisons revealed that there is a significant difference between day and nighttime LSTs, with nighttime LSTs from different satellite sensors more consistent than the daytime LSTs. This emphasizes the need to incorporate as much night-time LST data as available when predicting or optimizing fine-scale LSTs in the nighttime, so as to minimize the bias. The framework designed by this study will serve as a guideline towards efficient spatial optimization and harmonized use of LSTs when utilizing different satellite images associated with an array of land covers and at different times of the day.

**Keywords:** land surface temperature; mono-window algorithm; split-window algorithm; Landsat-8; Himawari-8





## 1. Introduction

Land surface temperature (LST) is widely adopted in a range of meteorological, hydrological, and ecological applications [1–3] for assessing variability in the Earth's climate system. Estimating spatial LST is primarily constrained when using the conventional ground monitoring stations under varying spatial and temporal changes [4,5]. Thus, the adoption of satellite remote sensing techniques has been used frequently for effective LST measurement [6,7]. Particularly, data retrieved from the thermal infrared (TIR) band of both sun-synchronized and geostationary satellites have been adopted widely for LST estimation around the globe [8,9].

However, there are different combinations of spatial and temporal resolutions of LSTs obtainable from satellite data due to technical limitations in the design of these satellites' TIR sensors [10,11]. This can be learned from the sun-synchronized sensors that provide

data with finer spatial resolution but low temporal resolution, while the spatial resolution of geostationary satellite data is relatively coarse but has a finer temporal resolution. This has led to the limited use of these satellite-derived LSTs, especially for diurnal LST analysis where high-spatial-resolution LSTs at different times of the day (daytime and nighttime) are required [2,12]. For example, among various sun-synchronized satellite platforms, with the advanced developments of high-resolution Thermal Infrared Sensor (TIRS) in Landsat-8, the sensor can potentially provide global-scale and high-quality LST data at 100 m spatial resolution, yet the temporal resolution is about 16 days. On the other hand, data from the TIRS of the NOAA Geostationary Operational Environmental Satellites (GOES) can provide LST data at a high temporal resolution ranging from 30 min to 1 h, but these data have a relatively low spatial resolution of 4 km. There are also several sun-synchronized satellites that produce LST data with moderate temporal and spatial resolution, e.g., Sentinel-3 and Moderate Resolution Imaging Spectroradiometer (MODIS) satellites' sensors, which can provide LST data at 1 km spatial resolution at a temporal resolution of 12 h.

To achieve high-spatial-resolution and high-temporal-resolution data suitable for diurnal analysis from satellite sensors, studies have focused on the harmonized use of data from different satellites and data optimization using fusion models. However, harmonized use of satellite data is still primarily focused on data in the visible spectrum of remote sensing satellites. Considering the systematic bias between the satellite sensors, their resolution differences and the discrepancies in LST inversion algorithms can affect the accuracy of their fusion or harmonize use [13]. With the blending spatiotemporal temperatures model (BLEST), integrating the image fusion and spatiotemporal fusion model (IIFSM) and spatial and temporal adaptive emissivity fusion model (STAEFM) having RMSEs of up to 0.6 K, 1.82 K, and 5.7 K, respectively [2,7,14]. It is important to understand the relationship between the satellite images to be fused and the resulting biases from the combination of different retrieval algorithms. For example, daytime LSTs can be retrieved from Landsat-8 using several algorithms such as the mono-window algorithm (MWA), split-window algorithm (SWA), and single-channel algorithm (SCA) [15]. Sentinel-3 Sea and Land Surface Temperature Radiometer (Sentinel-3 SLSTR) on the other end can provide both daytime and nighttime LST data at 1 km spatial resolution, given its dual-view scanning temperature radiometer, while the Himawari-8 satellite can provide LST data with a spatial resolution of 2 km in 10 min intervals. The harmonized use of LST images from Landsat-8, Sentinel-3, and Himawari-8 can potentially generate high-spatial-resolution LSTs both in daytime and nighttime. The accuracy of the results greatly depends on the compatibility of Landsat-8's LST retrieval algorithms with those of the other satellites [15]. However, existing cross-comparison studies have primarily focused on the visible and near-infrared band of satellite sensors [16,17], while cross-comparison between TIR bands generally receives less attention.

Despite very little research focusing on the cross-comparison of remotely sensed data in the TIR band, some are exceptionally worth noting. One study compared LST data retrieved from the TIRS of the Landsat-8 satellite with in situ LST measurements using a broadband thermal infrared radiometer on the island of Mallorca, Spain [18]. The three algorithms studied in the LST retrieval included the SCA, SWA, and radiative transfer equation (RTE). The results show that SWA obtained the lowest root-mean-squared error (RMSE) in a range between 1.6 K to 2 K. In comparison, the other study compared the LST generated from the MODIS with a medium resolution loaded on the sun-synchronized satellite with that retrieved from the data obtained by a geostationary satellite (GOES) [19]. The LST from the MODIS satellite was retrieved using SWA, and the dual-window algorithm (DWA) was used alternatively for the LST retrieval from the GOES satellite due to the lack of split-window channels in the GOES (12) satellite. It revealed that LST generated from the GOES satellite was higher overall when compared with that retrieved from the MODIS satellite. The temperature difference was also reported to be larger in the daytime when compared with the nighttime, which can be explained by anisotropy in satellite viewing geometry and land surface properties. In addition, Jee et al. compared LST data in Japan derived from

two sun-synchronized satellites, i.e., Landsat and MODIS. The study observed a positive correlation between Landsat and MODIS LST and a RMSE of 4.61 K [20]. However, when the LST from two satellites were compared with observations from automatic weather stations (AWS), a stronger correlation and a smaller RMSE error were obtained for both satellites (0.83 and 3.28 K) and MODIS (0.96 and 2.25 K). The difference between two LSTs was attributed to the difference in optical observation and variance of spatial resolution. The studies on cross-comparison of LST discussed above primarily focused on (i) identifying satellites with the highest accuracy for estimating LST in a specific locality, (ii) resolving differences between products from different satellites, and (iii) identifying the most precise LST retrieval algorithm for a particular satellite.

Given the potential to achieve high-spatial-resolution and high-temporal-resolution LST from the harmonized use or fusion of LST products from Landsat-8, Sentinel-3 SLSTR, and Himawari-8 TIRS, which will be suitable for diurnal analysis, this study aims to conduct a cross-comparison analysis on the LST products from these three satellites. The objectives of this study are to (i) develop a framework for LST comparison between the TIRS of various satellites; (ii) examine the effect of the differences in retrieval algorithms on the relationship between LSTs retrieved from different satellite sensors; and (iii) examine the relationship between remotely sensed LSTs during daytime and nighttime. In addition, a null hypothesis that there is no significant relationship between the LSTs retrieved from different satellite sensors will also be tested.

## 2. Study Area and Data

### 2.1. Study Area

Hong Kong is located at the entrance of Pearl River Delta of China (see Figure 1a), at latitude 22°9′14″N~22°33′44″N and longitude 111°50′7″E~114°26′30″E [21], with an approximate size of 1104.4 km$^2$. Hong Kong is dominated by several hills and highlands, resulting in a remarkably high development density in the few areas with relatively shallow slopes across its 18 districts (See Figure 1b) [22]. The intense urbanization in the region has consequently resulted in the development of an urban heat island (UHI) effect with large temperature differences between urban and rural areas, making the region a perfect site for cross-comparison of LST from different remote sensing satellites [23]. The study of LST in relation to spatial and temporal peculiarities of this region will help to understand the factors responsible for the vast variation in the temperature of urban and rural landscapes in the area. A population of about 7.5 million was recorded in 2020 and the region is considered as one of the world's most densely populated cities [24]. Hong Kong has a monsoon-influenced humid subtropical climate, according to the Köppen–Geiger climate classification [25], and its climate has received a lot of attention by local researchers because of the considerable increase in the occurrence of extreme hot weather events in the region in recent decades [26].

### 2.2. Data

#### 2.2.1. Landsat-8 Satellite Data

There are several sun-synchronized satellites hosted with TIR sensors [10]. In this study, high-quality data from the Landsat-8 satellite were adopted with high spatial resolution, availability of long-term series, and availability of multiple TIR bands (bands 10 and 11) [27]. Landsat-8 has a relatively fine spatial resolution in 30 m for both visible and near-infrared bands, and 100 m for TIR bands with a revisit period of every 16 days for all bands [28]. The Landsat-8 satellite carries the Operational Land Imager (OLI), which stores remotely sensed data in visible spectra into nine spectral bands; and the Thermal Infrared Sensors (TIRS), which feature two bands in the TIR region centered at the 10.9 μm and 12 μm atmospheric windows (see Figure 2) [16,29]. To carry out cross-comparison analysis for this study, cloud-free Landsat-8 data covering Hong Kong were sourced from the United States Geological Survey website [30] (Table 1).

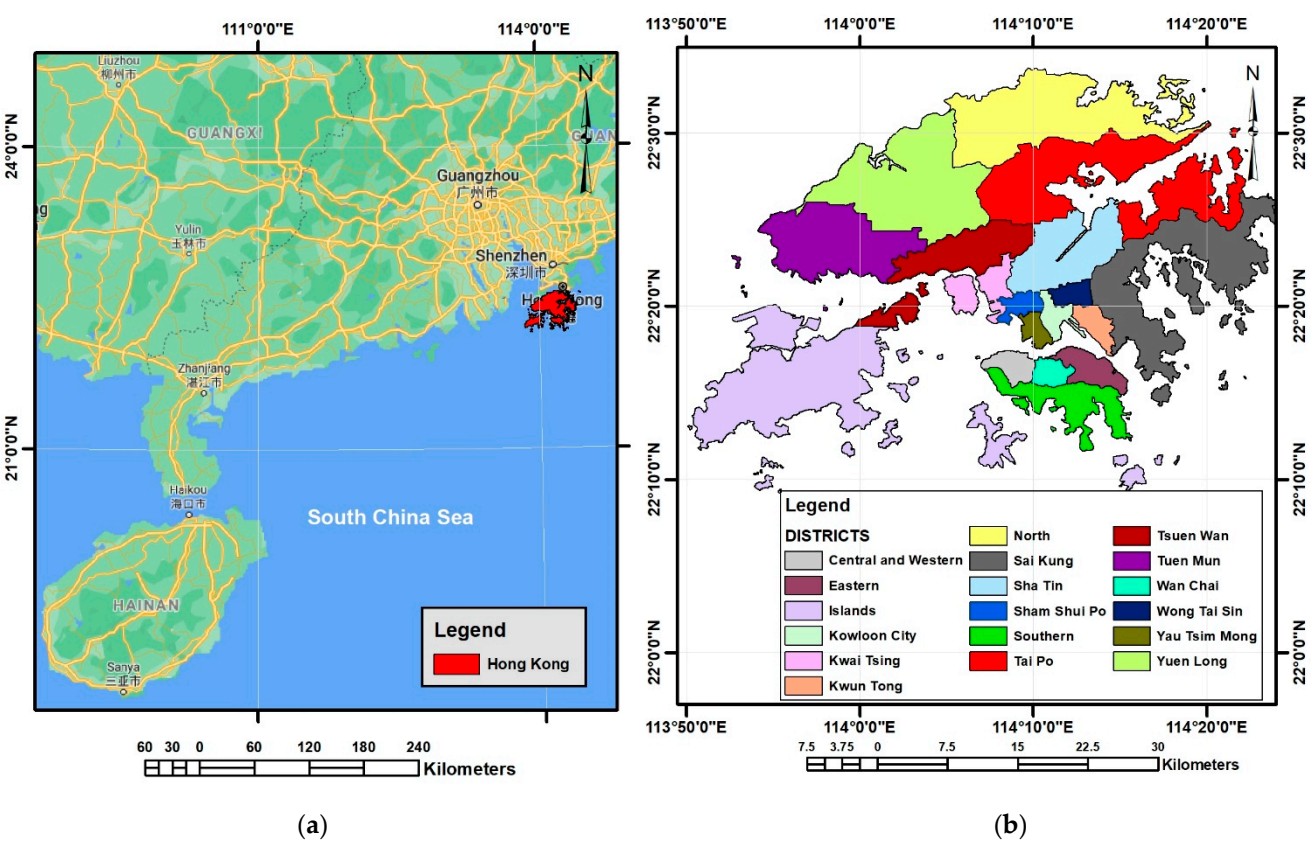

(**a**)                                                                    (**b**)

**Figure 1.** The study area in Hong Kong SAR. (**a**) Hong Kong in southeast China; (**b**) the map of Hong Kong with 18 districts.

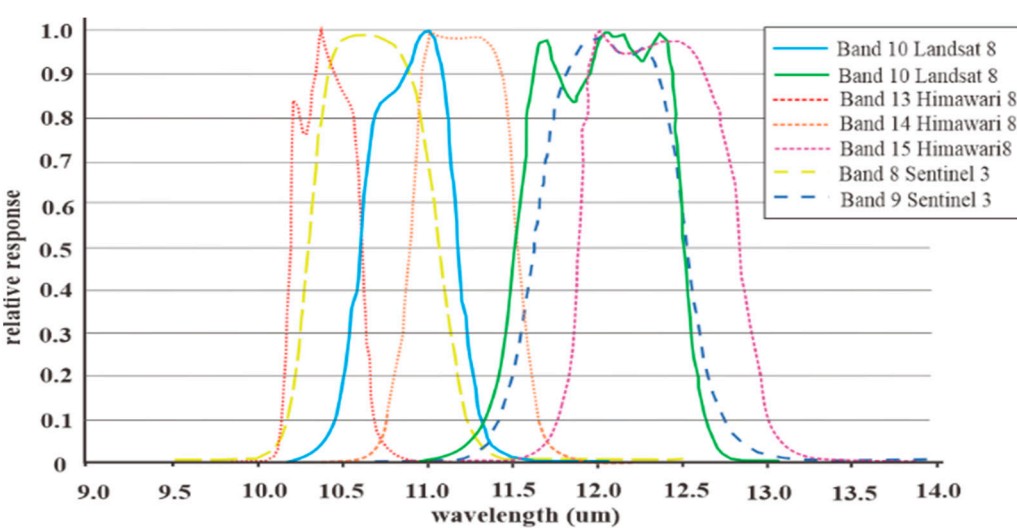

**Figure 2.** Spectral response of Landsat 8, Sentinel-3, and Himawari-8 TIRS thermal channels.

**Table 1.** Details of satellite data used for this study.

| Satellite Data | Date | Overpass Time | Period |
|---|---|---|---|
| Landsat 8 | 19 January 2021 | 10:52 am | daytime |
| Sentinel-3 SLSTR | 18 January 2021 | 10:48 pm | nighttime |
| | 19 January 2021 | 10:48 am | daytime |
| Himawari-8 | 18 January 2021 | 10:50 pm | nighttime |
| | 19 January 2021 | 10:50 am | daytime |

### 2.2.2. Sentinel-3 SLSTR Data

In order to complement LST data from Landsat 8 that can only be used to derive LST data in the daytime at 16-day intervals, Sentinel-3 SLSTR, which can produce daily LST data at daytime and nighttime, was also adapted in this study [31]. Like the Landsat-8 satellite, Sentinel-3 SLSTR is also a sun-synchronized satellite but with a moderate spatial resolution (1 km). The presence of two TIR bands centered at the 10.5 μm and 12.4 μm atmospheric windows can be used to derive LST products using SWA. The daytime and nighttime Sentinel-3 SLSTR data for Hong Kong were downloaded from the "Europe's eyes on Earth" website where it was archived [32].

### 2.2.3. Himawari-8 Satellite Data

Since several geostationary satellites are available, data from the Advance Himawari Imager (AHI) on the Himawari-8 Japanese metrological satellite were selected as a major data source for this study. The Himawari-8 satellite is a new-generation geostationary satellite that was launched by the Tanegashima Space Center [33]. This satellite was chosen for this study because of its improved TIR sensors and a better spatial (2 km) and temporal resolution (+10 min for full disk and 2 min for Japan) when compared with other geostationary satellites [33]. In addition, the presence of three TIR bands centered at 10.4 μm, 11.2 μm, and 12.4 μm can be used to derive LST products using SWA [29]. The Himawari-8 satellite provides a coverage of the entire East Asia and Western Pacific regions [34]. The AHI data for Hong Kong were downloaded from the JAXA website, where it was archived in the NetCDF format using the P-Tress system.

### 2.2.4. Land Use Data

Land use involves the modification and management of the natural environment into the built environment [35]. These modifications consequently influence land surface emissivity (LSE), which is a key input in estimating LST, making land use data indispensable for this study. The Land Utilization in Hong Kong (LUHK) obtained in December 2020 was used to extract land classification information of the study area [36]. The LUHK map was created utilizing updated satellite photos, the Planning Department's in-house survey data, and other pertinent data from various government ministries. The data have a spatial resolution of 10 m, covering the administrative boundary of Hong Kong. Thus, for this study, the LUHK data were collected and resampled by interpolating the values to the resolution of the satellite data. The collected LUHK map has 27 different land use classes and was further integrated into 10 classes by grouping land use classes with similar characteristics together [37]. The area under study is a combination of both vegetation, developed land, and water bodies, as presented in Figure 3 and Table 2.

**Table 2.** The percentage of each land use class in Hong Kong.

| S/N | LUHK Class (Abbreviation) | Percentage (%) |
|---|---|---|
| 1 | Residential (RES) | 7.082 |
| 2 | Commercial (COM) | 0.42 |
| 3 | Industrial (IND) | 2.40 |
| 4 | Institutional (INS) | 12.56 |
| 5 | Agricultural (AGR) | 4.41 |
| 6 | Green Space (GS) | 65.73 |
| 7 | Undeveloped (UND) | 1.78 |
| 8 | Waterbody (WB) | 4.21 |
| 9 | Others (OT) | 1.41 |
| | Total Land Surface coverage | 100.00 |

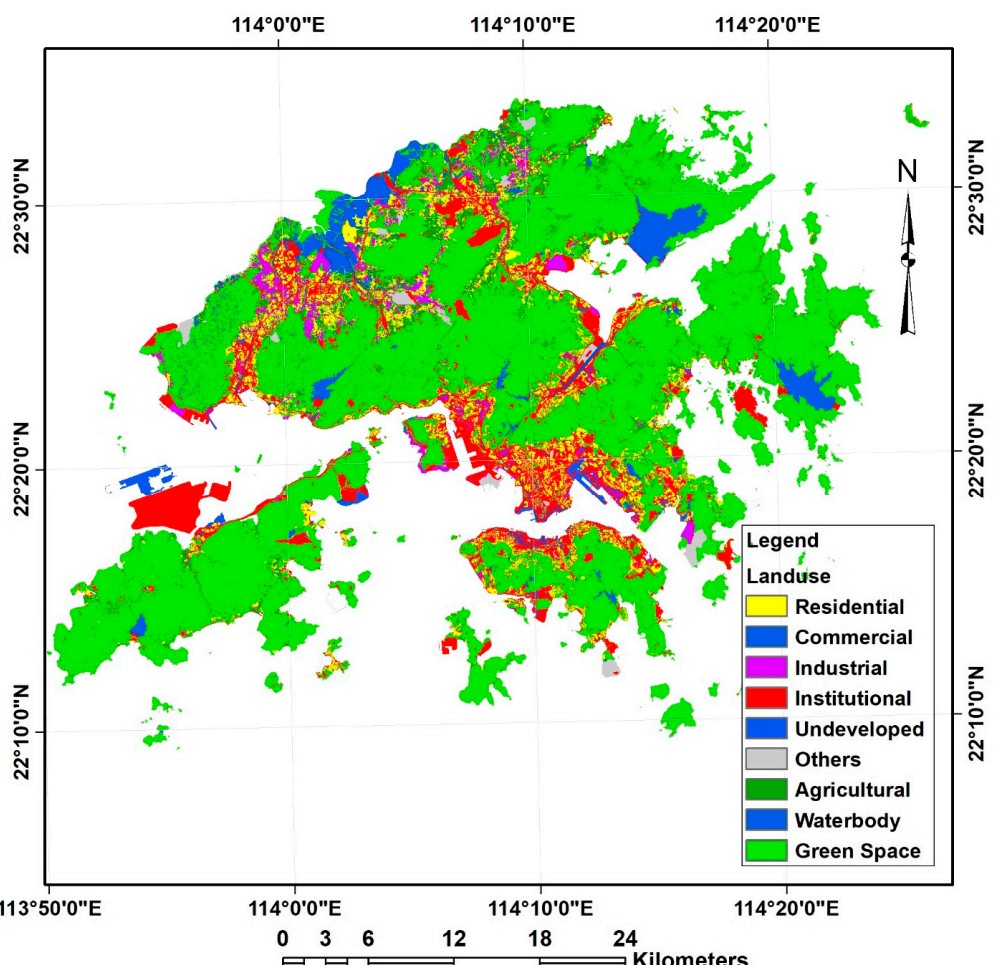

**Figure 3.** Hong Kong land cover map (LUHK 2020).

## 3. Methodology

The process involved in the cross-comparison analysis can be classified into two main tasks as presented in the framework in Figure 4, including (i) the retrieval of LST from satellite data and (ii) cross-comparison of the LST product obtained from the different satellite data.

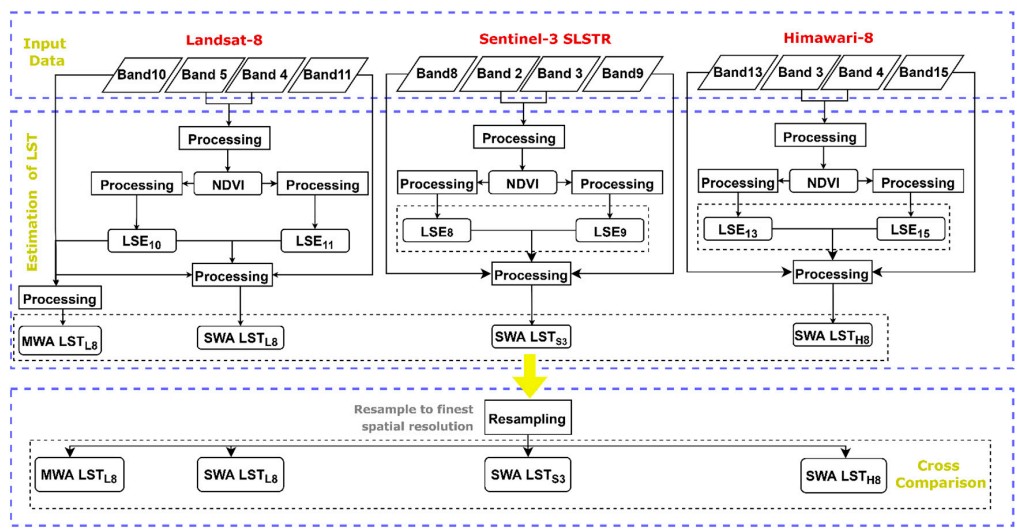

**Figure 4.** Framework for LST retrieval and comparison.

### 3.1. Estimation of LST from Satellite Data

The first step in the research design was to estimate the LST products from the satellites, and this was carried out by using existing retrieval algorithms. Compared to Sentinel-3 SLSTR and Himawari-8, there are a number of retrieval algorithms available for the retrieval of LST from Landsat-8. In this study, two widely used methods were employed to retrieve LST from Landsat-8 [26], i.e., the MWA and the SWA, while for Sentinel-3 SLSTR and Himawari-8, SWA was the only LST retrieval algorithm that is widely employed [31,36], which gives referencing and a baseline to further develop this study.

#### 3.1.1. LST Retrieval from Landsat-8

- Mono-Window Algorithm

In this study, the MWA was adopted to estimate LST using information from a single TIR band (band 10) of the Landsat-8 satellite [38]. The method, which has an accuracy of 2 K, requires two major parameters, i.e., brightness temperature (BT) and land surface emissivity (LSE) ($\varepsilon_\lambda$), to estimate LST [38]. The computation of the LST is presented in Equation (1):

$$LST = \frac{BT}{\{1 + [(\lambda BT/\rho)\ln\varepsilon_\lambda]\}} \tag{1}$$

where $BT$ represents the brightness temperature of the TIR band, $\lambda$ is the emitted radiance's wavelength (10.895), $\rho$ is the Boltzmann constant ($1.38 \times 10^{-23}$ J/K), and $\varepsilon$ is the land surface emissivity.

- Split-Window Algorithm

The SWA, also referred to as the multichannel algorithm, was used to retrieve LST from the Landsat-8 data. The algorithm depends on more than one TIR band for the retrieval of LST from satellite data. The algorithm chosen for this study has received a wide acceptance for LST retrieval with high accuracy (RMSE = 0.93 K) [39]. In addition, it does not require additional information about the atmospheric profile during satellite acquisition. Landsat-8 contains dual TIR bands (band 10 and 11), which makes it possible to retrieve LST by using SWA. This study used SWA [39] for the estimation of LST from Landsat 8 TIR data, as presented in Equations (2)–(4).

$$LST = T_{10} + B_1(T_{10} - T_{11}) + B_0 \tag{2}$$

$$B_0 = \frac{C_{11}(1 - A_{10} - C_{10})L_{10} - C_{10}(1 - A_{11} - C_{11})L_{11}}{C_{11}A_{10} - C_{10}A_{11}} \tag{3}$$

$$B_1 = \frac{C_{10}}{C_{11}A_{10} - C_{10}A_{11}} \tag{4}$$

$T_{10}$ and $T_{11}$ are the $BT$ of the Landsat-8 band 10 and band 11. $B_0$ and $B_1$ are coefficients in the algorithm, which can be calculated using the expressions in Equations (3) and (4). $A_{10}$, $A_{11}$, $C_{10}$, and $C_{11}$ in the equations are intermediate parameters determined by the LSE and atmospheric transmittance (AT) for band 10 and 11. The AT for the two Landsat-8 TIR bands can be estimated using water vapor, as presented in Table 3 [39].

**Table 3.** The relationship between atmospheric transmittance ($\tau10/11$) and water vapor content (w).

| Model | Water Vapor Range | Equation |
|---|---|---|
| Mid-latitude summer region | 0.2–3.0 g/cm$^2$ | $\tau_{10} = -0.0164w^2 - 0.04203w + 0.9715$ <br> $\tau_{11} = -0.01218w^2 - 0.07735w + 0.9603$ |

The water vapor content (w) in the equations in Table 3 for this study was estimated from the relative humidity (RH) and near-surface temperature ($T_o$) of the study area based on Equation (5) [40].

$$W_i = 0.0981 \times \left\{ 10 \times 0.6108 \times \exp\left[\frac{17.27 \times (T_o - 273.15)}{273.3 + (T_o - 273.15)}\right] \times RH \right\} + 0.1697 \quad (5)$$

After estimating the AT, $A_{10}$, $A_{11}$, $C_{10}$, and $C_{11}$ can then be calculated using Equations (6)–(9).

$$A_{10} = \varepsilon_{10}\,\tau_{10} \quad (6)$$

$$A_{11} = \varepsilon_{11}\,\tau_{11} \quad (7)$$

$$C_{10} = (1 - \tau_{10})\,[1 + (1 - \varepsilon_{10})\,\tau_{10}] \quad (8)$$

$$C_{11} = (1 - \tau_{11})\,[1 + (1 - \varepsilon_{11})\,\tau_{11}] \quad (9)$$

where $\tau_{10}$ and $\tau_{11}$ represent AT for bands 10 and 11, respectively; $\varepsilon_{10}$ and $\varepsilon_{11}$ represent the LSE for bands 10 and 11, respectively. Meanwhile, $L_{10}$ and $L_{11}$ were computed using the variables in Table 4 depending on the range of the BT of bands 10 and 11 [20]. From the table, A is the slope and B(K) is the linear regression intercept. Thus, if the BT of band 10 ranges between $-10$ and $20$ °C, the value of $L_{10}$ will be estimated as $0.4087 \times T_{10} - 55.58$. The same method was used for the estimation of $L_{11}$.

**Table 4.** Linear regression coefficients for $L_{10}$ and $L_{11}$.

| TIR Bands | T Range (°C) | A | B(K) |
|---|---|---|---|
| Band 10 | −10–20 | 0.4087 | −55.58 |
|  | 20–50 | 0.4464 | −66.61 |
| Band 11 | −10–20 | 0.4442 | −59.85 |
|  | 20–50 | 0.4831 | −71.23 |

- Calculating Land Surface Emissivity

The NDVI threshold method (NBEM) was adopted to estimate LSE in this study. This method is simple and accurate, such that it is more preferred when compared to other methods used in several studies [33,41,42]. Firstly, the NDVI is estimated from the satellite data's *red* and near-infrared (*NIR*) band of the satellite data, as presented in Equation (10), after which the LSE can be estimated.

$$NDVI = \frac{NIR - red}{NIR + red} \quad (10)$$

In order to consider the high signal-to-noise ratio (SNR), the NDVI threshold method adopted for this study considered the impact of different land use cover types, i.e., vegetation, bare soil, impervious surface, and water, in the estimation of LSE [43]. Pixels with NDVI values smaller than 0 were considered as water bodies, pixels with NDVI values greater than 0.50 were considered to be fully vegetated, while pixels with NDVI values smaller than 0.2 were considered as non-vegetated areas [41]. The ASTER spectral database was used to determine the emissivity value of the different land use cover types for both band 10 and band 11, as presented in Table 5 [41]. Finally, pixels with NDVI values that fall between the nonvegetated pixels (NDVI < 0.2) and fully vegetated pixels (NDVI > 0.5) were considered to be a mixed pixel. The LSE for these pixels was estimated using Equation (11) [41,44]:

$$\varepsilon_m = \varepsilon_1 Pv + \varepsilon_2 (1 - Pv) + (1 - \varepsilon_2)(1 - Pv)F\varepsilon_1 \quad (11)$$

where $\varepsilon_m$ is the emissivity of mixed pixels, $\varepsilon_1$ is the emissivity value for fully vegetated pixels, and $\varepsilon_2$ is the emissivity value for nonwater and nonvegetated pixel. *F* is a geometrical factor whose value ranges between 0 to 1 depending on the surface [45]. For this study, the value was defined as equaling to 0.05. $P_v$ in this study was estimated using Equation (12):

$$P_V = \left(\frac{NDVI - NDVI_S}{NDVI_V - NDVI_S}\right)^2 \quad (12)$$

where *NDVI* is the result of estimation from Equation (10); $NDVI_v$ and $NDVI_s$ represent the *NDVI* values for vegetation and sand, respectively, which are quantified as 0.5 and 0.2, accordingly [44].

**Table 5.** The emissivity values of water, vegetation, and nonvegetation for Landsat-8 TIRS band 10 and band 11.

|  | $\varepsilon$ **for Water** | $\varepsilon$ **for Vegetation** | $\varepsilon$ **for Non-Vegetation** |
|---|---|---|---|
| TIR—band 10 | 0.991 | 0.984 | 0.964 |
| TIR—band 11 | 0.986 | 0.980 | 0.970 |

### 3.1.2. LST Retrieval from Sentinel-3 SLSTR

Given that Sentinel-3 SLSTR data can provide LST data both during daytime (~11:00 am) and nighttime (~11:00 pm), LST was retrieved at both time instants using SWA for the study area. Although surface temperature can be retrieved from Sentinel-3 SLSTR data using the dual-angle algorithm (DAA) as well, the choice of SWA was due to its better performance over land surface when compared with DAA, due to the differences in the footprints and the observation geometries between the two views used in the DAA [31]. The Zheng19 SWA algorithm, which has an accuracy of 1.3 K, was adopted in this study for both daytime and nighttime LST retrieval from Sentinel-3 SLSTR [46]. This algorithm was adopted because it is refined from the generalized SWA (Equation (2)) that has been adaptive to the spectral channel of Sentinel-3 SLSTR. Equation (13) describes the Zheng19 algorithm [46]:

$$LST = d_0 + \left( d_1 + d_2 \frac{1-\varepsilon}{\varepsilon} + d_3 \frac{\Delta\varepsilon}{\varepsilon^2} \right) \frac{T_8 + T_9}{2} + \left( d_4 + d_5 \frac{1-\varepsilon}{\varepsilon} + d_6 \frac{\Delta\varepsilon}{\varepsilon^2} \right) \frac{T_8 + T_9}{2} \\ + d_7 (T_8 - T_9)^2 \tag{13}$$

where $d_n$ are equation constants, as presented in Table 6 (*n* = 0–7); $T_8$ and $T_9$ are input data that represent the BT of the Sentinel-3 SLSTR band 8 and band 9, respectively; while $\Delta\varepsilon$ is the difference between the emissivity of the two TIR channels, i.e., $\Delta\varepsilon = (\varepsilon_8 - \varepsilon_9)$; and $\varepsilon$ represents the average emissivity of the two TIR channels ($\varepsilon = 0.5 \times (\varepsilon_8 + \varepsilon_9)$).

**Table 6.** Coefficients of SWA used for retrieving LST from Sentinel-3 SLSTR.

| Coefficient | Value |
|---|---|
| $d_0$ | −0.51 |
| $d_1$ | −0.053 |
| $d_2$ | −0.180 |
| $d_3$ | 2.13 |
| $d_4$ | 0.377 |
| $d_5$ | 71.4 |
| $d_6$ | −10.04 |
| $d_7$ | −5.9 |

The two input variables for the daytime and nighttime estimation of moderate resolution LST from Sentinel-3 SLSTR as presented in Equation (13) are the BT and LSE of the two TIR channels. Since the TIR band of Sentinel-3 SLSTR contains the information of BT, the only input parameter required to be estimated was the LSE and it was retrieved using the NDVI threshold-based method, following the expressions in Equation (14):

$$\varepsilon_\lambda = \begin{cases} a_\lambda + b_\lambda \rho_{red} & NDVI < NDVI_s \\ \varepsilon_{v\lambda} P_v + \varepsilon_{s\lambda}(1 - Pv) + C_\lambda & NDVI_s \le NDVI \le NDVI_v \\ \varepsilon_{v\lambda} + C_\lambda = 0.99 & NDVI > NDVI_v \end{cases} \tag{14}$$

where $C_\lambda$ is the cavity effect of the thermal band, and the subscripts "s" and "v" stand for soil and vegetation, respectively. Meanwhile, $Pv$ is the percentage of vegetation and $\rho_{red}$ is the surface reflectance of the red band of the Sentinel-3 STSLR data. For the estimation of daytime LSE, the *NDVI* and $P_V$ were estimated following Equations (10) and (12) respectively, with the data in Sentinel-3 SLSRT band 2 (S2) and band 3 (S3) representing the red and NIR band, respectively. Considering that the satellites cannot collect data in the visible band (S2 and S3) at nighttime, LSE value estimated during daytime was also employed for the LST estimation at nighttime, assuming that there will be no significant change in the LSE within the same day.

### 3.1.3. LST Retrieval from Himawari-8

The algorithm developed by Choi and Shu [33] was employed in this study owing to high retrieval accuracy (RMSE = 1.083 K), efficiency, and similarity to the one used for the Landsat data and Sentinel-3 SLSTR by making use of two TIR bands (bands 13 and 15). The equation for the LST retrieval is presented as below.

$$LST = c_0 + c_1 BT_{13} + c_2(BT_{13} - BT_{15}) + c_3(\sec\theta - 1) + c_4(1 - \bar\varepsilon) + c_5\Delta\varepsilon \qquad (15)$$

where $c_1$ to $c_5$ are Himawari-8 LST retrieval coefficients presented in Table 7, and $\Delta\varepsilon$, and $\bar\varepsilon$ are the difference in LSE and mean LSE across the two TIR bands. $BT_{13}$ and $BT_{15}$ are the BT of bands 13 and 15, respectively, and $\theta$ is the viewing zenith angle (VZA) of the Himawari-8 data, which was estimated based on the locational properties (longitude and latitude) of the data [47]. Given that Himawari-8 has a temporal resolution of 10 min for the full disc, which includes the study area, LST was estimated both during the day and at nighttime for the detailed comparison with the moderate-resolution LST retrieved from Sentinel-3 SLSTR.

**Table 7.** Coefficients of the algorithm used for retrieving LST from Himawari-8 data.

|  | Conditions | $C_0$ | $C_1$ | $C_2$ | $C_3$ | $C_4$ | $C_5$ |
|---|---|---|---|---|---|---|---|
| | Moist | 67.1857 | 0.7448 | 2.07 | 1.096 | 63.061 | −75.1606 |
| Day | Normal | 8.926 | 0.9651 | 0.9364 | −0.1385 | 56.8638 | −63.8708 |
| | Dry | 15.3567 | 0.9461 | 1.1996 | −1.411 | 48.5137 | −68.3093 |
| | Moist | 44.5826 | 0.8205 | 2.0427 | 1.6411 | 58.5399 | −59.1371 |
| Night | Normal | 12.1778 | 0.9535 | 0.9278 | −0.095 | 51.2696 | −51.8349 |
| | Dry | 20.3004 | 0.9279 | 1.0879 | −1.4883 | 47.2503 | −61.7212 |

The NDVI threshold method was also adopted for the LSE inversion for Himawari-8 using Equation (16) [28], with band 3 and band 4 of the Himawari-8 data representing the *red* and *NIR* band, respectively. $\varepsilon_{v\lambda}$ and $\varepsilon_{g\lambda}$ are the equation constants for the vegetation emissivity and ground emissivity, respectively and were adopted from the study of [28] for different land-use classes, while the emissivity constant for the water body was estimated as the average of the total emissivity of all the land use classes in the study area. In addition, the cavity effect ($C_i$) for urban areas was estimated based on the Himawari-8 Satellite Zenit Angle (SZA), as shown in Table 8 [28], to reduce the emissivity error as proposed by Atitar and Sobrino [48]. Similarly to the estimation of LST from Sentinel-3 SLSTR at nighttime, it was also assumed that LSE in the study area remains constant throughout the day, so the LSE data estimated during the daytime could be used in the LST estimation at nighttime.

**Table 8.** The cavity effect ($C_i$) in the urban area for the three TIR bands.

| Band | SZA (°) | | | | | | |
|---|---|---|---|---|---|---|---|
| | 0 | 10 | 20 | 30 | 40 | 50 | 60 |
| 13 | 0.0104 | 0.0115 | 0.0125 | 0.0136 | 0.0147 | 0.0155 | 0.0161 |
| 14 | 0.0104 | 0.0109 | 0.0114 | 0.0119 | 0.0124 | 0.0128 | 0.0131 |
| 15 | 0.0089 | 0.0092 | 0.0096 | 0.0099 | 0.0102 | 0.0106 | 0.0108 |

*3.2. Cross-Comparison of LSTs from Satellite Data*

Spatial and statistical analysis were conducted to assess the relationship between the daytime (~11:00 am) LSTs from the Landsat-8 satellite using both MWA and SWA and the LSTs from Sentinel-3 SLSTR and Himawari-8 satellites retrieved using SWA. Given that Landsat-8 data are not generally available at nighttime, only the relationship between LSTs from Sentinel-3 STSLR and Himawari-8 satellite data was assessed at nighttime (~11:00 pm).

LST data from Landsat-8 derived using MWA and SWA ($MWA_{L8}$ and $SWA_{L8}$) were compared against the LST data from Sentinel-3 SLSTR ($SWA_{S3}$) and Himawari-8 ($SWA_{H8}$) to investigate the bias in their combined use and identify the combination that has the least bias. The relationship between Sentinel-3 and Himawari-8 both during the daytime and nighttime was also assessed to estimate bias resulting from their combined use.

According to Wu et al. [49], when a moderate- or coarse-resolution LST image has been resampled to the same spatial resolution as the fine-resolution LST image, the relationship between the temperature of the pure homogeneous coarse-resolution pixel, which is covered by only one land cover (LC) type, and the corresponding fine-resolution pixel, can be described by a linear equation expressed as:

$$LST_{F(x,y,dn)} = LST_{C(x,y,\,dn)} + R \tag{16}$$

where $LST_F$ and $LST_C$ represent the LST of the fine resolution and resampled moderate or coarse resolution, respectively. Meanwhile, $(x, y)$ is the pixel location for both fine- and coarse-resolution LST; $dn$ is the acquisition date; and $R$ is the bias, which is the difference between the LST observed at the fine and moderate or coarse resolutions.

Since LST images obtained from Landsat-8, Sentinel-3 SLSTR, and Himawari-8 have a spatial resolution of 100 m, 1000 m, and 2000 m, respectively, Sentinel-3 SLSTR and Himawari-8 were also resampled to 100 m by using the bilinear resampling method. The estimated LSTs from three satellites were also projected to the same coordinate system. In addition, the processed land use data (LUHK) of the study area were then used to extract LSTs by land use type from the satellite data.

Following Equation (16), the relationship between the moderate-resolution LST data ($SWA_{S3}$) and the corresponding fine-resolution LST ($MWA_{L8}$ and $SWA_{L8}$) for each land use class can be expressed as:

$$MWA_{L8(LU,dn)} = SWA_{S3(LU,dn)} + MWA_{L8-S3} \tag{17}$$

$$SWA_{L8(LU,dn)} = SWA_{S3(LU,dn)} + SWA_{L8-S3} \tag{18}$$

where $MWA_{L8}$ and $SWA_{L8}$ are the average LST values for the land use class as measured from the Landsat-8 data using MWA and SWA, respectively; $SWA_{S3}$ is the LST value as measured from the Sentinel-3 SLSTR data; $LU$ is the land use class; $MWA_{L8-S3}$ and $SWA_{L8-S3}$ are the biases that can be estimated by subtracting $SWA_{S3}$ from $MWA_{L8}$ in Equation (17) or $SWA_{L8}$ in Equation (18), respectively. Meanwhile, $dn$ denotes the acquisition date of the satellite data. Similarly, the relationship between the coarse-resolution LST from $SWA_{H8}$ and the corresponding fine-resolution LST from $MWA_{L8}$ and $SWA_{L8}$ for each land use class following Equations (17) and (18) can be expressed as presented in Equations (19) and (20), respectively.

$$MWA_{L8(LU,dn)} = SWA_{H8(LU,dn)} + MWA_{L8-H8} \tag{19}$$

$$SWA_{L8(LU,dn)} = SWA_{H8(LU,dn)} + SWA_{L8-H8} \tag{20}$$

where $SWA_{H8}$ is the LST value as measured from the Himawari-8 data, $MWA_{L8-H8}$ and $SWA_{L8-H8}$ are the biases, which can be estimated by subtracting $SWA_{H8}$ from $MWA_{L8}$ in Equation (19) or $SWA_{L8}$ in Equation (20), respectively.

For comparisons between coarse ($SWA_{H8}$)- and moderate ($SWA_{S3}$)-resolution LSTs, the coarse-resolution LST data (2000 m) were resampled to the same resolution as moderate-

resolution data (1000 m). The two data were subsequently projected to the same coordinate system together with the land use data. The resulting data were then compared using a similar Equation as the daytime.

$$SWA_{S3(LU,dn)} = SWA_{H8(LU,dn)} + SWA_{S3-H8} \tag{21}$$

where $SWA_{S3}$ and $SWA_{H8}$ represent both the daytime and nighttime LST values as measured from Sentinel-3 SLSTR and Himawari-8 satellites, respectively, and $SWA_{S3-H8}$ is the relationship bias, which can be estimated by subtracting $SWA_{S3}$ from $SWA_{H8}$ in Equation (21). The RMSE, SD, and the average of the biases of the land use classes were further estimated.

## 4. Results

### 4.1. Comparison of Remotely Sensed LSTs Retrieved Using Different Retrieval Algoritims

Figure 5 reveals that LST maps generated from $SWA_{H8}$ (Figure 5a), $SWA_{S3}$ (Figure 5b), and from Landsat-8 using both MWA (Figure 5c) and SWA (Figure 5d) all had a similar pattern for Hong Kong on 19 January 2021. Overall, water bodies have lower LST values, while the developed land has higher LST values. However, the LST map generated from Sentinel-3 SLSTR ($SWA_{S3}$) and Himawari-8 ($SWA_{H8}$) could not provide more detailed heterogeneity of temperature distribution compared with that generated from Landsat-8 ($MWA_{L8}$ and $SWA_{L8}$) satellite data. That is due to the difference in resolution between the satellite sensors, with Landsat-8 having a finer spatial resolution (100 m). The temperature range of the results from $SWA_{H8}$ is the smallest (280 K to 293 K), followed by $SWA_{S3}$ (279 K to 294 K), $MWA_{L8}$ (276 K to 304 K), and $SWA_{L8}$ (274 K to 312 K), which has the widest temperature range.

The average LST by land use class on 19 January 2021, as presented in Table 9, reveals that as estimated from all satellite data, commercial (COM) land use has the lowest average LST, with an average temperature of 287.88 K for $MWA_{L8}$, 289.53 K for $SWA_{S3}$, 290.4 for $SWA_{H8}$, and 290.04 K for $SWA_{L8}$. In contrast, the land use class with the highest average LST is industrial (IND) land use, based on the estimations from all the satellite data, with an average temperature of 289.76 K for $MWA_{L8}$, 290.68 K for $SWA_{S3}$, 290.98 for $SWA_{H8}$, and 291.42 K for $SWA_{L8}$, respectively. Similarly, the other land use class (OT), which comprise rocky shores, cemeteries, and construction sites, also have high LSTs for $SWA_{H8}$, $SWA_{S3}$, and the two retrieval algorithms for Landsat-8, while the agricultural (AGR) and green space (GS) land use classes have a relatively low average LST as estimated from the three satellites, demonstrating that the green vegetation cover had a considerable cooling effect.

**Table 9.** Relationship between average LST estimated from Himawari-8 ($SWA_{H8}$), Sentinel-3 SLSTR ($SWA_{S3}$), and Landsat-8 ($SWA_{L8}$ and $MWA_{L8}$) on 19 January 2021 by land use class.

| LUHK Class | $MWA_{L8}$ (K) | $SWA_{S3}$ (K) | $SWA_{H8}$ (K) | $MWA_{L8-S3}$ ($\Delta K$) | $MWA_{L8-H8}$ ($\Delta K$) | $SWA_{L8}$ (K) | $SWA_{L8-S3}$ ($\Delta K$) | $SWA_{L8-H8}$ ($\Delta K$) |
|---|---|---|---|---|---|---|---|---|
| RES | 288.26 | 289.96 | 290.41 | −1.70 | −2.15 | 289.5 | −0.46 | −0.91 |
| COM | 287.88 | 289.53 | 290.4 | −1.65 | −2.52 | 289.04 | −0.49 | −1.36 |
| IND | 289.76 | 290.68 | 290.98 | −0.92 | −1.22 | 291.42 | 0.34 | 0.24 |
| AGR | 288.68 | 290.01 | 290.33 | −1.33 | −1.65 | 290.08 | 0.07 | −0.25 |
| INS | 288.77 | 289.8 | 289.86 | −1.03 | −1.09 | 290.46 | 0.26 | 0.2 |
| GS | 288.21 | 289.91 | 290.03 | −1.70 | −1.82 | 289.95 | 0.04 | −0.08 |
| UND | 287.1 | 288.9 | 289.55 | −1.8 | −2.45 | 289.02 | 0.12 | −0.53 |
| OT | 289.12 | 290 | 290.65 | −0.88 | −1.53 | 290.65 | 0.65 | 0 |
| Bias (K) | | | | −1.38 | −1.80 | | 0.77 | −1.46 |
| SD (K) | | | | 0.39 | 0.54 | | 0.49 | 0.65 |
| RMSE (K) | | | | 1.20 | 1.66 | | 0.87 | 1.04 |

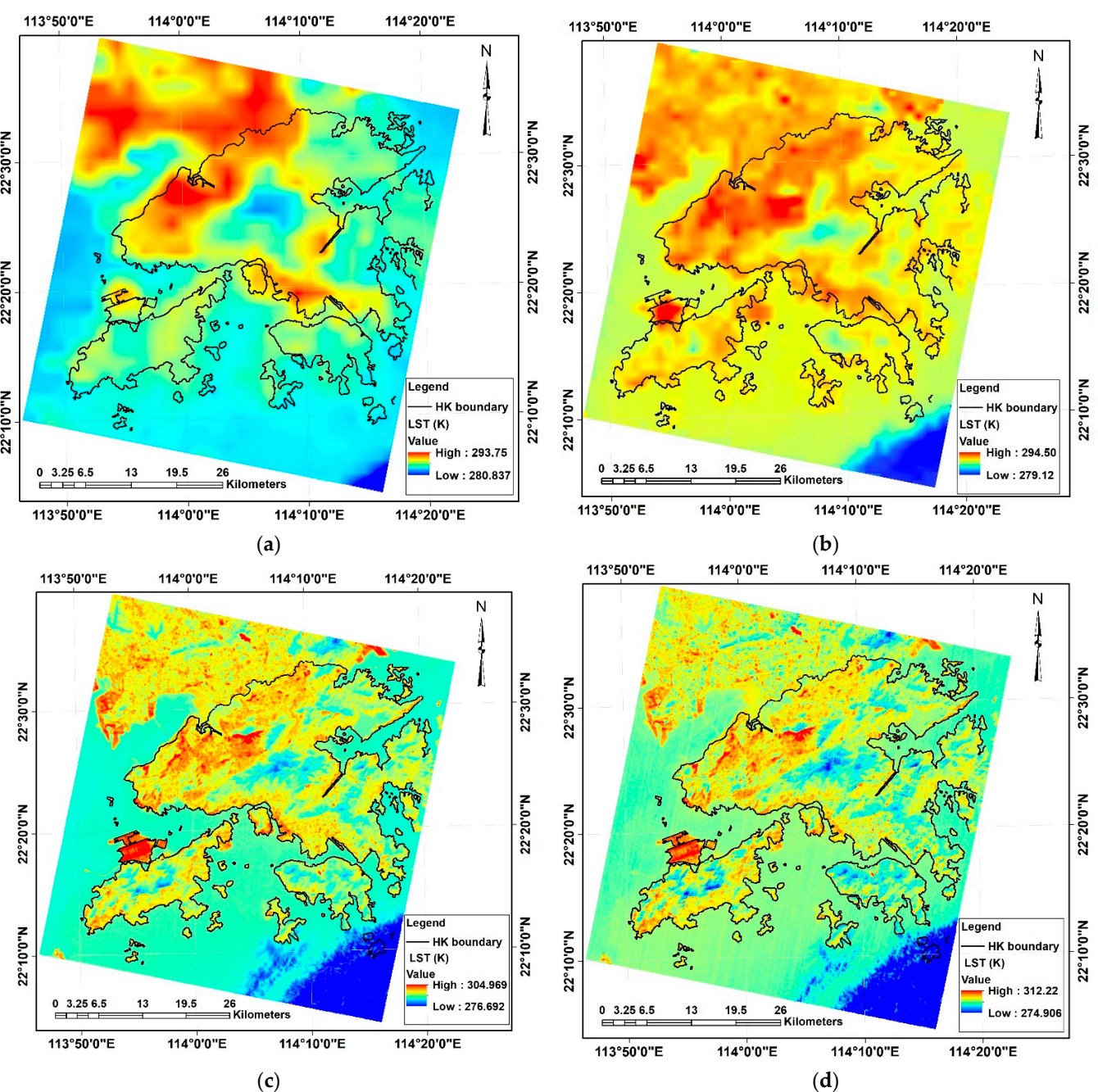

**Figure 5.** LST map of Hong Kong on 19 January 2021. (**a**) LST retrieved from SWA$_{H8}$; (**b**) LST retrieved from MWA$_{S3}$; (**c**) LST retrieved from MWA$_{L8}$; (**d**) LST retrieved from SWA$_{L8}$.

However, from the Landsat-8 TIR sensor, MWA$_{L8}$ was relatively lower and dominated with negative bias in all land use, while SWA$_{L8}$ was relatively higher and largely dominated with positive bias in all the land use classes when compared with moderate-resolution LST (SWA$_{S3}$), but largely dominated with negative bias when compared with coarse-resolution LST (SWA$_{H8}$). This variation is due to the differences in the inversion algorithm.

For MWA $_{L8-S3}$, the result indicated that the estimated LSTs obtained from Landsat-8 are lower than those from Sentinel-3, with the undeveloped (UND) land use class largely underestimated in having the largest bias (−1.8 K). Meanwhile, for MWA$_{L8-H8}$, the greatest underestimation was found in the commercial (COM) land use class, with a bias of −2.52 K. For SWA$_{L8-S3}$, the overestimation (the estimated LST from Sentinel-3 is lower than that from Landsat-8) was largest for the other (OT) land use class, with a positive bias of 0.65 K; while

the least bias was 0.04 K, recorded in the area covered by green space (GS) land use class. This finding agrees with the study of Yu et al. [41] that an overestimation was recorded for LST estimated from Landsat using SWA when compared with in situ measurements, and an underestimation was recorded when MWA was used for the LST estimation from Landsat data.

The RMSE, as presented in Table 9, is 1.20 K and 1.66 K when $SWA_{S3}$ and coarse $SWA_{H8}$ are compared with fine-resolution LST from Landsat-8 using MWA, while the RMSE of $SWA_{L8-S3}$ and $SWA_{L8-H8}$ is 0.87 K and 1.04 K, respectively. This reveals that LST retrieved from Landsat-8 using SWA is more consistent with LST retrieved from the moderate and coarse-resolution LST.

It can also be deduced from the statistical analysis as presented in Table 9 that moderate-resolution LST ($SWA_{S3}$) is more closely related to the fine-resolution LST when compared with the relationship between the coarse-resolution ($SWA_{H8}$) and fine-resolution LSTs. The close relationship between moderate- and fine-resolution LST is further revealed in Figure 6, where the boxplots of the LSTs estimated from the three satellites (Sentinel-3 SLSTR, Himawari-8, and Landsat-8 using both MWA and SWA) revealed that the first quartile of $SWA_{S3}$, as well as the third quartile and maximum value, are more consistent with the values from the Landsat-8 satellite ($MWA_{L8}$ and $SWA_{L8}$). However, $SWA_{S3}$ is more consistent with the $SWA_{L8}$ estimation than $MWA_{L8}$.

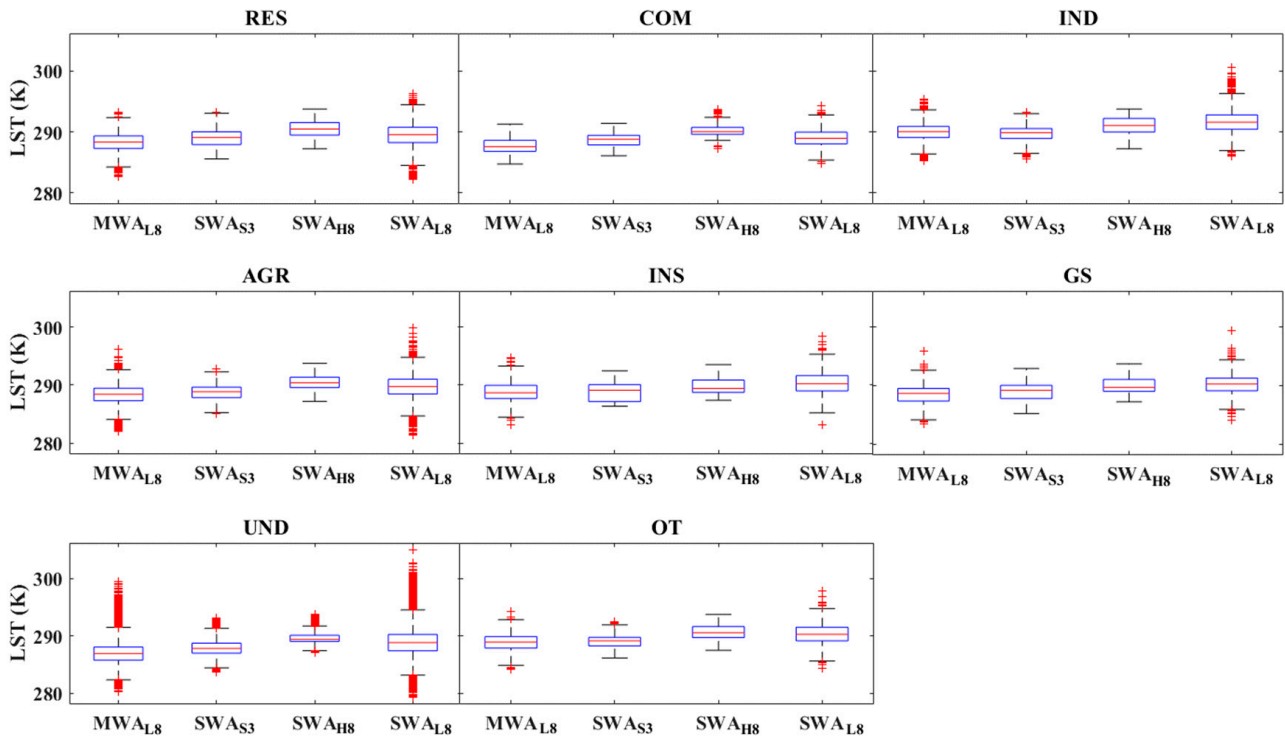

**Figure 6.** Boxplots of LSTs estimated from Himawari-8, Sentinel-3, and Landsat-8 using MWA and SWA for the different land use classes during winter.

The regression analysis of the LSTs presented in Figure 7 further depicts the high variability in the relationship between LST data from different satellite sensors when the varying retrieval algorithm is employed. The lowest values are reported in Figure 7a,c, while Figure 7b,d show higher correlation values. This difference also suggests that moderate- and fine-resolution LST is more closely related as compared with the relationship between the coarse- and fine-resolution LST.

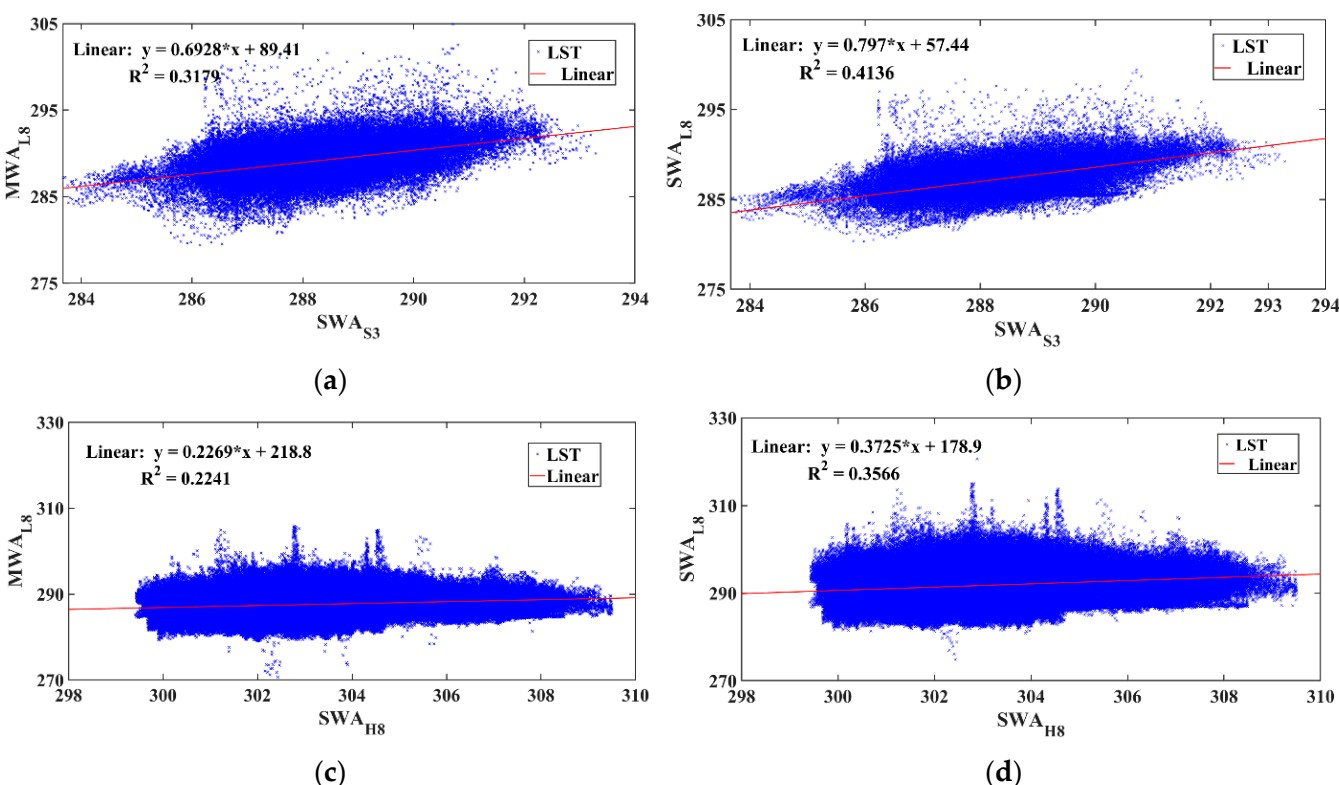

**Figure 7.** Four regressions of the average LSTs in the daytime. (**a**) Regression of LSTs obtained from $SWA_{S3}$ and $MWA_{L8}$. (**b**) Regression of LSTs obtained from $SWA_{S3}$ and $SWA_{L8}$. (**c**) Regression of LSTs obtained from $SWA_{H8}$ and $MWA_{L8}$. (**d**) Regression of LSTs obtained from $SWA_{H8}$ and $MWA_{L8}$.

### 4.2. Cross-Comparison between Remotely Sensed LST during Daytime and Nightime

This study generated both nighttime (~11:00 p.m.) and daytime (~11:00 a.m.) LST maps from $SWA_{H8}$ (Figure 8a,c) and $SWA_{S3}$ (Figure 8b,d) on 18 January 2021 and 19 January 2021, respectively. Similar to the observation in daytime, the LST maps from the two satellite exhibited a similar temperature distribution. However, in contrast to the observation in the daytime, water bodies are generally characterized with high LST value at nighttime together with developed land in the study area. The LST maps also revealed that temperatures for all land use classes are relatively lower in the nighttime than in the daytime, and this can be related to the absence of shortwave radiation from the sun at nighttime. Thus, LST measurements are solely based on longwave radiation emitted from the land surface. Since Himawari-8 TIRS has a coarser spatial resolution, the range of LSTs (280.114 K–286.281 K) from Himawari-8 is smaller than that from Sentinel-3 SLSTR (278.937 K–287.937 K).

The average LST from $SWA_{S3}$ and $SWA_{H8}$ during the day (~11:00 a.m.) and nighttime (~11:00 p.m.) by land use class, presented in Table 10, further confirmed the inference from the LST map, as the average LST for all land use classes during the daytime is higher than the estimate at nighttime, with the industrial (IND) land use class having the largest temperature difference as estimated from both $SWA_{S3}$ (7.93 K) and $SWA_{H8}$ (8.39 K). This is closely followed by the agricultural (AGR), green space (GS), and residential (RES) land use classes, which also have large temperature differences between daytime and nighttime as estimated from both satellites. In contrast, the commercial (COM) land use class has the lowest temperature difference when the daytime and nighttime estimated LSTs from the two satellites are compared (i.e., 4.56 K and 6.16 K from $SWA_{S3}$ and $SWA_{H8}$, respectively).

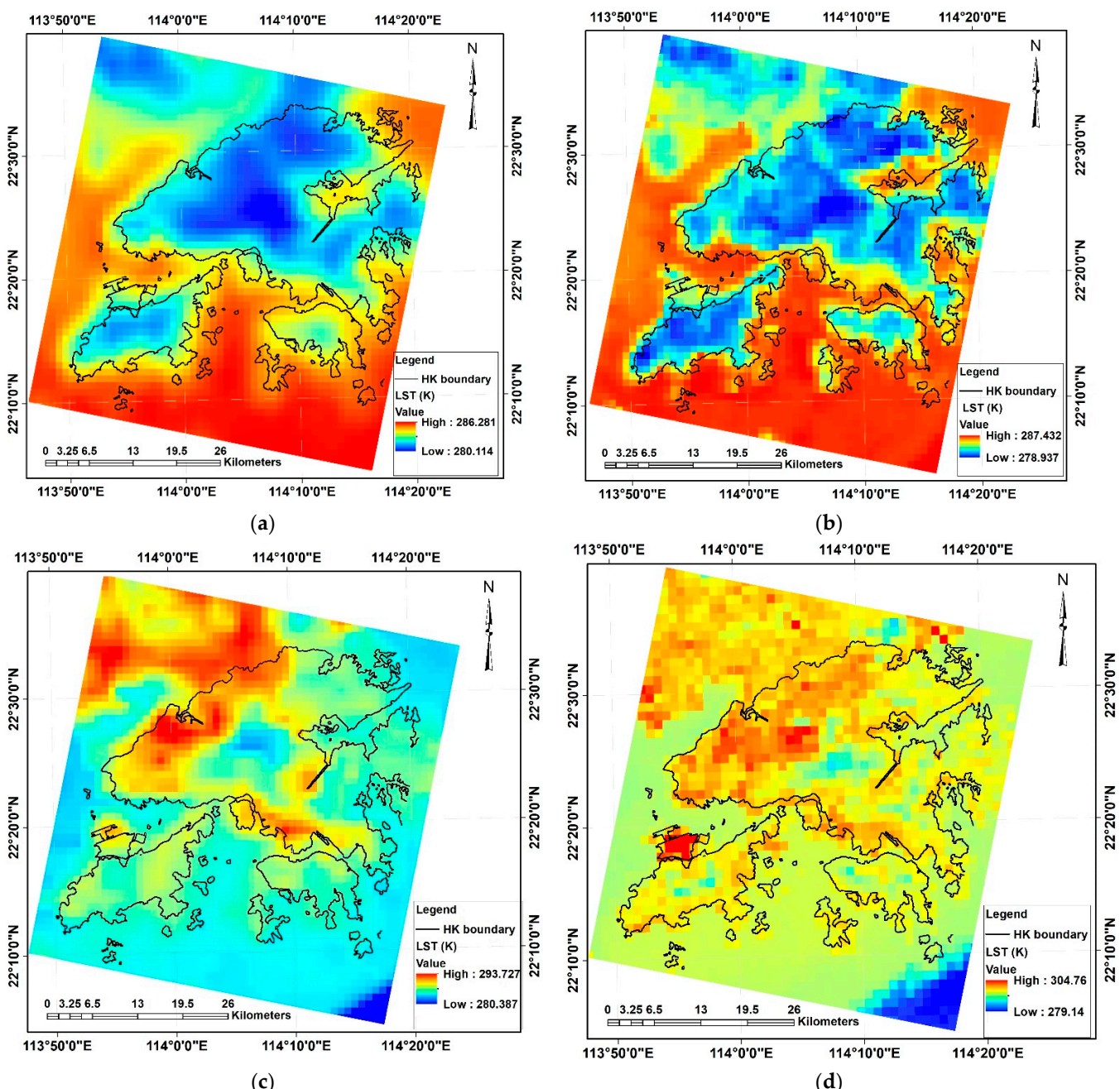

**Figure 8.** LST maps of Hong Kong on 18 January 2021 at nighttime (around 11 pm). (**a**) LST retrieved from SWA$_{H8}$. (**b**) LST retrieved from SWA$_{S3}$. (**c**) LST retrieved from MWA$_{L8}$ (**d**) LST retrieved from SWA$_{L8}$.

When comparing the relationship between LST from moderate (SWA$_{S8}$)- and coarse (SWA$_{H8}$)-resolution satellites at nighttime (~11:00 p.m.), estimates from SWA$_{S3}$ are overestimated with positive bias in six of the eight land use classes. Meanwhile, during the daytime(~11:00 a.m.), the moderate-resolution LST is generally dominated by negative bias in all land use classes. However, the bias is insignificant both during the nighttime and daytime, ranging between −0.47 to 0.73 and −0.87 to 0.06, respectively, with the relationship in the nighttime more significant, with an RMSE of 0.35 K. The greater RMSE in the daytime (0.38) could be due to the effects of LSE and viewing geometry.

**Table 10.** Relationship between average LST from SWA$_{H8}$ and SWA$_{H8}$ during the daytime (~11:00 a.m.) and nighttime (~11:00 p.m.) by land use class.

| LUHK Class | Nighttime | | | | Daytime | | | Daytime–Nighttime | |
|---|---|---|---|---|---|---|---|---|---|
| | SWA$_{S3}$ (K) | SWA$_{H8}$ (K) | SWA$_{S3-H8}$ (ΔK) | SWA$_{S3}$ (K) | SWA$_{H8}$ (K) | SWA$_{S3-H8}$ (ΔK) | SWA$_{S3}$ (ΔK) | SWA$_{-H8}$ (ΔK) |
| RES | 283.27 | 282.77 | 0.5 | 289.96 | 290.41 | −0.45 | 6.69 | 7.64 |
| COM | 284.97 | 284.24 | 0.73 | 289.53 | 290.4 | −0.87 | 4.56 | 6.16 |
| IND | 282.75 | 282.59 | 0.16 | 290.68 | 290.98 | −0.3 | 8.2 | 8.66 |
| AGR | 281.81 | 281.67 | 0.14 | 290.01 | 290.33 | −0.32 | 7.93 | 8.39 |
| INS | 283.12 | 282.9 | 0.22 | 289.8 | 289.86 | −0.06 | 6.68 | 6.96 |
| GS | 282.46 | 282.48 | −0.02 | 289.91 | 290.03 | −0.12 | 7.45 | 7.55 |
| UND | 283.46 | 283.25 | 0.21 | 288.9 | 289.55 | −0.65 | 5.44 | 6.3 |
| OT | 282.59 | 283.06 | −0.47 | 290 | 290.65 | −0.65 | 7.41 | 7.59 |
| Bias (K) | | | 0.18 | | | −0.43 | 6.8 | 7.41 |
| SD (K) | | | 0.38 | | | 0.50 | 6.90 | 7.45 |
| RMSE (K) | | | 0.35 | | | 0.38 | 1.25 | 0.90 |

The regression analysis of the LSTs presented in Figure 9 further depict a closer relationship between LSTs retrieved from moderate and coarse resolution in the nighttime with a regression coefficient of 0.61, as compared to the relationship in the daytime.

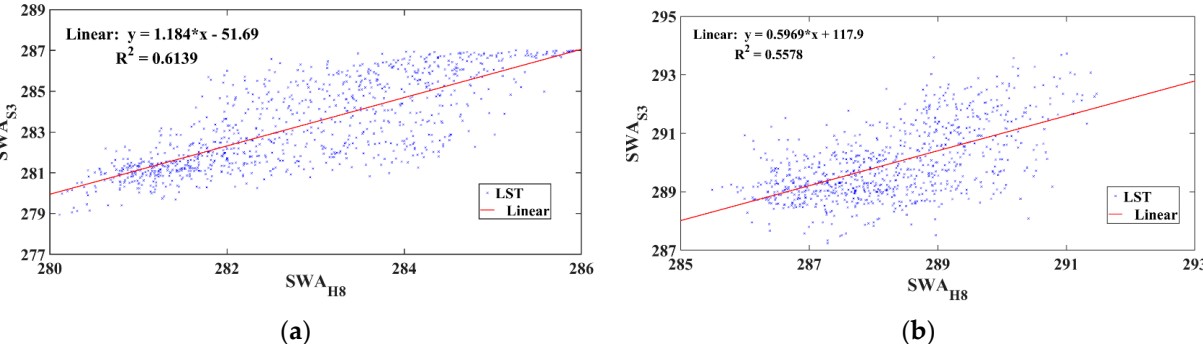

**(a)**

**(b)**

**Figure 9.** Regressions of the average LSTs in the daytime and nighttime. (**a**) Regression of LSTs obtained from SWA$_{S3}$ and SWA$_{H8}$ at nighttime on January 18, 2021 (~11:00 p.m.); (**b**) regression of LSTs obtained from SWA$_{S3}$ and SWA$_{H8}$ at daytime on 19 January 2021 (~11:00 a.m.).

## 5. Discussion

This study developed a framework for the cross-comparison of LST data retrieved from sun-synchronized and geostationary satellites using different algorithms, and we compared the biases resulting from the relationship between LSTs both during the daytime and nighttime. In order to achieve high-spatial-resolution and high-temporal-resolution data from satellite sensors, studies on image fusion and harmonized use of satellite data leveraged on the relationship between sun-synchronized and geostationary satellite data [14,50,51]. However, the bias in the relationship influenced the accuracy of the resulting data [2,6,14]. Given that there are several techniques for LST retrieval from satellite data, the first step for achieving an optimal result from harmonized use of LST data or image fusion is to identify the combination of retrieval algorithm for sun-synchronized and geostationary data to achieve the minimum bias. In order to achieve this, our framework recommends the retrieval of LST from both the sun-synchronized and geostationary satellite using different retrieval methods, depending on the availability of data, after which the retrieved LST data from the sun-synchronized and geostationary data can be compared based on land use class to identify the combination that will result in the least bias.

The choice of land use class as the basis for comparison is borne out of the entangled relationship between LST and land use class as established in different literatures [52,53]. Findings from the comparison of LSTs retrieved from Sentinel-3 SLSTR and Himawari-8

satellites using SWA with LSTs retrieved from Landsat-8 using both MWA and SWA revealed similarity in the spatial distribution pattern of resulting LSTs, of the LST maps from the three satellites irrespective of the retrieval algorithm. However, as suggested in the literature [54,55] and confirmed by this study, LSTs derived from the three satellite sensors are not directly compactible. The disparities in estimated LST between Himawari-8, Sentinel-3 SLSTR, and Landsat 8 are directly related to variances in the spectral bandwidth and radiometric resolution between the satellite sensors (see Figure 2). The difference in spectral band resolution also affects the relationship between the LSTs from three sensors, and in particular, the dynamic range of LST. LST retrieved from the Sentinel-3 SLSTR satellite has a closer relationship with LST retrieved from Landsat-8 both using MWA and SWA when compared with the relationship between LSTs from Himawari-8 and Landsat 8. This is because of the closer spatial resolution range between Sentinel-3 SLSTR and Himawari-8 (1000–100 m) when compared with the range of Himawari-8 and Landsat-8 (2000–100 m).

Further comparison of retrieved LSTs in the study revealed that there is a significant difference in the biases generated based on the retrieval algorithm. The biases of LSTs retrieved using similar algorithms ($SWA_{H8}$, $SWA_{S3}$ and $SWA_{L8}$) are comparable and insignificant (0.17 K and $-0.0.26$ K when $SWA_{L8}$ is compared with $SWA_{S3}$ and $SWA_{H8}$, respectively), while the biases are large when different algorithms are used ($SWA_{H8}$, $SWA_{S3}$, and $MWA_{L8}$). The reason is that when SWA was used for the retrieval for both satellite images, the estimation was based on the difference in the atmospheric absorbance of two adjacent thermal bands in a satellite (e.g., bands 10 and 11 for Landsat-8 and bands 13 and 15 for Himawari-8) and some environmental variables have also been considered, including relative humidity and water vapor, which ultimately reduces the temperature difference in the LST images from both satellites [38,39]. This implies that fusion or harmonized use of LSTs retrieved from sun-synchronized and geostationary satellites using similar algorithms is preferable because there will be less bias to account for. However, the pattern of the LST map generated from the Himawari-8 and Sentinel-3 SLSTR satellites are similar both in the daytime and nighttime, with daytime temperature generally being hotter [1]. The study demonstrates a higher correlation between $SWA_{H8}$ and $SWA_{S3}$ in the nighttime compared to daytime. The bias is less significant at nighttime, largely due to the lower dependency on differential surface cooling/heating at nighttime, and this makes nighttime LST more efficient for algorithm testing and temperature analysis [54]. In addition, SD during the daytime is relatively larger than at night. This can be linked with the impacts of structural shading, evaporative cooling, and changes in surface-air temperature [55]. Differential surface heating over various surface covers, such as trees and grass/soil, is another factor that contributes to the ample variation in LST in the daytime [1].

Finally, the study reveals that there is significant difference in the relationship between LSTs from different satellite sensors based on the retrieval algorithm employed, and also that there are huge discrepancies in the variation of LSTs based on the time of the day.

## 6. Conclusions

The need for high-spatial-resolution and high-temporal-resolution remotely sensed LST, which leads to the fusion and harmonized use of data from different satellites, has essentially provided a valuable insight to optimize the accuracy of LST. This study therefore proposed a method for comparing LST data retrieved from different satellite sensors with distinct retrieval algorithms. The proposed cross-comparison method is based on the land use classification of the study area. In this study, LST data for Hong Kong retrieved from Landsat-8 using both MWA and SWA were compared with those retrieved from the Sentinel-3 SLSTR and Himawari-8 satellites using SWA. A comparison was also carried out between daytime and nighttime LST retrieved from the Himawari-8 and Sentinel-3 SLSTR satellites. A significant difference was found between the LST images retrieved from the different satellites. The magnitude of the bias was revealed to be largely dependent on the attribute of the satellite sensor, retrieval algorithm, and the land use classes in the study

area. The comparison based on the time of the day revealed that nighttime LSTs from the two satellites (SWAH8 and SWAS3) are more consistent compared to daytime LSTs. This explains why LST studies that include assessment of the LST retrieval algorithm, accuracy assessment of LSTs from satellite sensors, and urban heat island analysis encourage the use of nighttime LSTs over LSTs retrieved in the daytime. Furthermore, considering the close relationship between LSTs from Himawari-8 and Sentinel-3 as observed in this study ($R^2 = 0.56$ and 0.61 for daytime and nighttime respectively), the combined use of satellite data from these sensors to provide high-spatial-resolution and high-temporal-resolution LSTs for diurnal LST analysis should also be investigated.

**Author Contributions:** Conceptualization: I.A.A. and M.S.W.; analysis: I.A.A. and J.Y.; writing— original draft preparation: I.A.A., R.Z. and M.S.W.; writing–review and editing: I.A.A., R.Z., J.Y., X.Z. and M.S.W.; funding acquisition, M.S.W. All authors have read and agreed to the published version of the manuscript.

**Funding:** This research was partly supported in part by the General Research Fund (project ID 15609421 and 15603920), the Collaborative Research Fund (Grant No. C5062-21GF), and the Hong Kong Ph.D. Fellowship Scheme from the Research Grants Council of Hong Kong. M. S. Wong thanks the support from the Research Institute for Land and Space (project ID 1-CD81), the Hong Kong Polytechnic University, Hong Kong, China.

**Data Availability Statement:** The data presented in this study are available on request from the corresponding author.

**Acknowledgments:** The authors would like to acknowledge the US Geological Survey (USGS), the Copernicus Open Access Hub European Space Agency (ESA), and Japan Meteorological Agency (JMA) for providing the Landsat-8 (OLI and TIR), Sentinel-3 SLSTR, and Himawari-8 archive. Furthermore, the authors would like to thank the reviewers and editors for their time and valuable input to this manuscript.

**Conflicts of Interest:** The authors declare no conflict of interest.

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
