# Peer review of "Cross-Comparison between Sun-Synchronized and Geostationary Satellite-Derived Land Surface Temperature: A Case Study in Hong Kong"

_remotesensing, doi:10.3390/rs14184444_

Round 1
Reviewer 1 Report (Previous Reviewer 2)
This is second time that I review the article untitled "Cross-comparison between Sun-synchronized and Geo-stationary satellite derived land surface temperature: A case study in Hong Kong". This article is interesting. The authors did improve the manuscript. According to me, it could be published.
Author Response
No comment to reply to from the reviewer
Reviewer 2 Report (New Reviewer)
Cross-comparison of LSTs retrieved from sun-synchronized and geostationary satellite data is an interesting topic, and is significant for applications of satellite derived LST. However, this manuscript fails to present this work with significant results, discussion and conclusions. There are many mistakes in both contents and english writing to make the paper lack readability. Therefore, the present manuscript dose not satisfy the conditions to be published. I recommend a rejection on it.

Author Response
Please see the attachment

Reviewer 3 Report (New Reviewer)
The ms remotesensing-1837333 with the title of Cross-comparison between Sun-synchronized and Geo-stationary satellite derived land surface temperature: A case study in Hong Kong has to be improved before it can go for further step.
L32-34 authors should revise the conclusion at the end of the abstract, and avoid using common words.
Make first characters small instead of capital for different words at same sentence, see L14 and 15.
L64 use h instead of hours
The introduction is very long, authors should make it shorter and remove invaluable text.
Please add the hypothesis at the end of the introduction.
Cite uncited text in the introduction, it is important issue.
Results were well presented in text, Tables and Figures, and I recommend the authors move some of these Tables of Figures into supplementary files.
The discussion has to be improved, it is short and there is no enough citation for other investigations. The authors should make it deeper than the current version.
References, please correct the mistakes and follow the guidelines of the journal. There are many mistakes, it seems that the authors took copy-paste references from other sources.
Regards, Reviewer
Round 2
Reviewer 2 Report (New Reviewer)
With major revision, the readability of this new version manuscript is improved. However, there are still many mistakes. Since it is a revised version, the tables, equations, and figures are changed comparing with the original version. Though i have listed some mistakes in the comments, the author should check through the entire manuscript very carefully for several times to avoid any mistakes again before the paper being accepted. As a result, i recommend a minor revision before the manuscript being published. Detailed comments can be found in the attachment.

Author Response
Please see the attachment

Reviewer 3 Report (New Reviewer)
The Ms has been improved
Author Response
No comment by the reviewer
This manuscript is a resubmission of an earlier submission. The following is a list of the peer review reports and author responses from that submission.
Round 1
Reviewer 1 Report
The present version of the paper is the result of a first review made by three reviewers. From my viewpoint, the authors have answered partially to the different questions raised by the reviewers, and there are still some additional important issues that should be addressed. In my opinion, the paper does not accomplish still with the high quality standards of remote sensing.
Although some modifications, authors still do not address the concerns in the present version. The sources of difference in LST results include sensor difference, inversion algorithm difference and resolution difference. The author tries to explain the differences as the relationship among resolution, urban heterogeneity and class, but it is obviously not well demonstrated. In other words, it is not enough to compare the results without explaining the reasons.
In image fusion, the overall overestimation or underestimation is easy to eliminate. However, this paper discusses this result more. This is of little help to readers or follow-up research. Give some useful suggestions or analysis.
Reviewer 2 Report
This is the second round of the article untitled « Cross-comparison between Landsat-8 and Himawari-8 satellite derived land surface temperature for image fusion: A case study 3 in Hong Kong”. The authors have fulfilled the previous recommendations. Then, after some minor advices that would follow, I think this article could be considered for publication in Remote Sensing MDPI:
§ 2.1: please add the Köppen-Geiger classification of Hong Kong
Fig. 2: has to be redone concerning the LULC. Please use the GLCNMO2013 to display the different LULC, has described in table 2. In addition, the scale can be improved using standard units (5 km, 10 km, and so on instead of 3,5 km and 7 km).
Round 2
Reviewer 1 Report
It is undeniable that the authors have done some work, mainly including: 1) inversion of LSTs from Landsat and Himawari data; 2) resampling to 100 m by using a bilinear resampling method; 3) Cross-comparison of resampled LSTs with different types in HongKong; and others. If the conclusion of this manuscript only refers to the comparison of LSTs from Landsat and Himawari at a specific spatial resolution, i think there is no problem. However, there are many descriptions and conclusions are aimed at image fusion, so there is a distance between the work and the conclusion.